# Position: Vector Prompt Interfaces Should Be Exposed to Enable Customization of Large Language Models

**Liangwei Yang** [1]  **Shiyu Wang** [1]  **Haolin Chen** [1]  **Rithesh Murthy** [1]  **Ming Zhu** [1]  **Jielin Qiu** [1]  **Zixiang Chen** [1]
**Juntao Tan** [1]  **Jianguo Zhang** [1]  **Zhiwei Liu** [1]  **Wenting Zhao** [1]  **Silvio Savarese** [1]  **Caiming Xiong** [1]  **Huan Wang** [1]
**Shelby Heinecke** [1]

## Abstract

As large language models (LLMs) transition from research prototypes to real-world systems, customization has emerged as a central bottleneck. While text prompts can already customize LLM behavior, we argue that text-only prompting does not constitute a suitable control interface for scalable, stable, and inference-only customization. This position paper argues that model providers should expose *vector prompt inputs* as part of the public interface for customizing LLMs. We support this position with diagnostic evidence showing that vector prompt tuning continues to improve with increasing supervision whereas text-based prompt optimization saturates early, and that vector prompts exhibit dense, global attention patterns indicative of a distinct control mechanism. We further discuss why inference-only customization is increasingly important under realistic deployment constraints, and why exposing vector prompts need not fundamentally increase model leakage risk under a standard black-box threat model. We conclude with a call to action for the community to rethink prompt interfaces as a core component of LLM customization.

## 1. Introduction

Large language models (LLMs) are increasingly deployed within an ecosystem in which model providers offer general-purpose inference APIs (Achiam et al., 2023; Comanici et al., 2025), while downstream application developers adapt these models to domain-specific products and services (Liang et al., 2025). In this setting, customization (Wu et al., 2025; Ding et al., 2023; Hu et al., 2022) serves as a

critical bridge between broadly capable foundation models and the heterogeneous requirements of real-world applications. Despite substantial advances in model capabilities, effective enterprise level deployment has fallen short of expectations. A central reason is that current interfaces provide limited support for systematic customization: in practice, downstream users primarily interact with LLMs through text-based prompts (Yuksekgonul et al., 2024; Fernando et al., 2023), which have become the de facto mechanism for shaping model behaviors.

Some model providers also offer fine-tuning interfaces to enable task-specific adaptation. However, within the prevailing division of responsibilities in the LLM ecosystem, fine-tuning is poorly aligned with the customization needs of downstream enterprises. Fine-tuning typically entails substantial computational and operational overhead, involves long iteration cycles, and affords limited granularity of control once a model is deployed. These properties make it difficult to support the large number of rapidly evolving tasks encountered in real-world applications, particularly when downstream users lack model ownership, training infrastructure, or the ability to frequently redeploy updated models. Consequently, neither text-based prompting nor fine-tuning alone provides a satisfactory foundation for scalable, inference-only customization in practice.

Although text-based prompts can and do customize LLM behavior in practice (Xiang et al., 2025; Martins et al., 2023), their effectiveness does not imply suitability as a system-level customization interface. Text prompts are discrete, semantically grounded artifacts whose influence on model behavior is mediated through linguistic interpretation, making them inherently brittle under iterative optimization (Shin et al., 2020; Yuksekgonul et al., 2024) and difficult to scale across heterogeneous tasks (Reynolds & McDonell, 2021). More importantly, treating customization primarily as prompt engineering (Ye et al., 2024) conflates expressiveness with controllability: while text prompts are expressive, expressiveness alone does not provide the stability, granularity, or learnability required for systematic adaptation under inference-only constraints. These limita-

[1]Salesforce AI Research, Palo Alto, CA, United States. Correspondence to: Liangwei Yang <liangwei.yang@salesforce.com>.

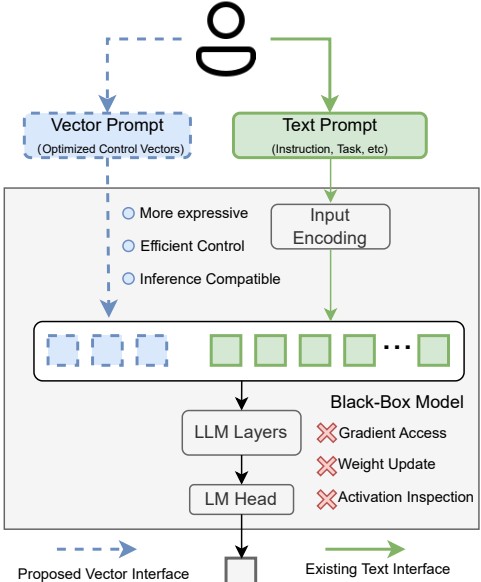

*Figure 1.* Illustration of prompt interfaces under black-box LLM deployment. Current LLM services predominantly expose text prompts as the customization interface. We argue that vendors should additionally expose vector prompts—optimized control vectors injected at the input-encoding stage—as a complementary interface. Vector prompt interfaces provide a more expressive and control-efficient parameterization while remaining compatible with inference-only access, without requiring gradient access, weight updates, or internal activation inspection.

tions suggest that customization should be reframed not as the manual design of textual instructions, but as a control interface design problem—one that requires interfaces capable of absorbing supervision, supporting iterative refinement, and remaining effective as task requirements evolve.

From this perspective, vector prompts provide a natural complement to text-only prompting as a customization interface. Unlike discrete textual instructions, vector prompts operate as continuous control signals that directly condition model computation (Liu et al., 2022), enabling more stable absorption of supervision and finer-grained adjustment of behavior under inference-only access (Sun et al., 2022a). Importantly, vector prompts need not be viewed as an optimization technique tied to a specific black-box training procedure, but rather as an interface abstraction that decouples customization from model weights while remaining compatible with black-box deployment settings. Figure 1 illustrates our position: exposing vector prompt inputs adds a complementary control interface to current text-based prompting under black-box deployment constraints.

**Position:** Model providers should expose vector prompt inputs as part of the public interface for customizing large language models. Elevating vector prompts, model providers can better support systematic, scalable, and inference-only customization for downstream applications without requiring task-specific tuning or model modification.

Finally, we clarify the scope of this position. We do not propose a new optimization algorithm for prompt tuning, nor do we claim that vector prompts universally outperform text-based prompts across all tasks. We also do not argue that fine-tuning should be eliminated as a model adaptation mechanism, or that text prompts are ineffective for guiding LLM behavior. Rather, our focus is on the design of customization interfaces under realistic deployment constraints. Specifically, we argue that exposing vector prompt inputs enables a more appropriate interface abstraction for scalable, systematic, and inference-only customization in black-box settings, independent of the particular optimization methods used to obtain such vectors.

## 2. Background

### 2.1. Prompting as a Control Interface

Prompting is commonly understood as providing natural language instructions to guide model behavior (Yuksekgonul et al., 2024; Murthy et al., 2025; Khattab et al., 2024; 2022). While this interpretation captures the surface form of interaction, it obscures the role that prompts play in deployed systems. From a systems perspective, prompts function not merely as linguistic instructions, but as control signals that condition model computation and influence downstream behavior. This distinction is critical when considering customization under realistic deployment constraints.

We use the term *interface abstraction* to refer to the form and semantics of control inputs exposed by a model for customization, independent of how such inputs are obtained or optimized. Viewed through this lens, a prompt specifies how users are allowed to interact with and adapt a model, defining the granularity, stability, and learnability of customization. An interface abstraction therefore determines what kinds of supervision can be absorbed, how reliably behavior can be adjusted, and whether customization can be carried out systematically over time.

Text prompts represent a particular interface abstraction in which control is mediated through discrete, semantically grounded tokens (Pryzant et al., 2023; Frick et al.; Hu et al.). This abstraction is accessible, but it also tightly couples customization to linguistic interpretation. As a result, the effectiveness of text prompts depends on semantic coherence, phrasing choices, and contextual interactions, which complicates iterative refinement and limits scalability across heterogeneous tasks. In contrast, alternative interface abstractions may expose control signals that are less constrained by natural language semantics, enabling different trade-offs between expressiveness, controllability, and stability.

| Customization Category | Representative Methods | Parameter Update | Gradient Access | Inference-only | Interface Layer |
|---|---|:---:|:---:|:---:|:---:|
| *Parameter-level customization (gradient-based)* | | | | | |
| Full Fine-tuning | SFT, DPO | ✓ | ✓ | ✗ | Parameters |
| Lightweight Fine-tuning | LoRA, Adapters | ✓(partial) | ✓ | ✗ | Parameters |
| Prompt Tuning | Prompt tuning, Prefix tuning | ✗ | ✓ | ✗ | Vector |
| *Prompt-based customization (no parameter update)* | | | | | |
| Manual Prompting | Hand-crafted text prompts | ✗ | ✗ | ✓ | Text |
| Text Prompt Optimization | TextGrad, RLPrompt | ✗ | ✗ | ✓ | Text |
| *Inference-only vector-based customization (black-box)* | | | | | |
| Black-box Vector Prompt Optimization | BBT, ZOO | ✗ | ✗ | ✓ | Vector |

*Table 1.* Organization of large language model customization methods by optimization access and interface layer. Representative methods include SFT and DPO (Rafailov et al., 2023); LoRA (Hu et al., 2022) and Adapters (Hu et al., 2023); Prompt tuning (Liu et al., 2022) and Prefix tuning (Li & Liang, 2021); TextGrad (Yuksekgonul et al., 2024) and RLPrompt (Deng et al., 2022); BBT (Sun et al., 2022b) and ZOO (Hu et al., 2024). Methods progress from gradient-based parameter adaptation to inference-only vector-based control, highlighting the trade-off between optimization access and deployment flexibility.

Framing prompting as a control interface, rather than solely as instruction following, shifts the focus of customization from manual prompt design to interface design. This perspective highlights that limitations commonly attributed to prompt engineering may instead arise from the interface abstraction itself, motivating the exploration of alternative forms of control inputs better suited for systematic and inference-only customization. Related systems also optimize the interface by rewriting or editing user prompts before model execution, including prompt transformation pipelines in generative media models such as MovieGen (team, 2024). Such approaches improve text-based interfaces by modifying the prompt before inference, whereas our position concerns exposing an additional non-text control interface for customization.

## 2.2. Text vs. Vector Prompts: A Conceptual Distinction

Text and vector prompts represent two distinct forms of control signals for customizing model behavior. The distinction between them is not merely one of representation format, but of the space in which control is expressed and the kinds of adaptation it enables. Understanding this distinction is essential for evaluating prompting mechanisms as interface abstractions rather than as isolated techniques.

Text prompts are instantiated as discrete token sequences and rely on natural language semantics to influence model behavior (Yuksekgonul et al., 2024; Shin et al., 2020; Wang et al., 2026). As an interface abstraction, text prompts tightly couple control to linguistic interpretation: the effect of a text prompt depends on phrasing, syntax, and contextual coherence, and its influence is mediated through the model's language understanding capabilities. This coupling makes text prompts expressive and human-readable, but also constrains customization to the discrete and semantically grounded structure of text.

In contrast, vector prompts operate as continuous vectors

that directly condition model computation (Liu et al., 2022; Sun et al., 2022a). Rather than encoding control through explicit linguistic meaning, vector prompts influence behavior through their position in a continuous representation space. As a result, they are less constrained by discrete tokenization and semantic coherence, and can absorb supervision in a more direct and fine-grained manner (Sun et al., 2022b). From an interface perspective, this enables smoother adjustment of behavior and more stable accumulation of task-specific information over iterative refinement.

These differences imply distinct trade-offs in controllability and scalability. Text prompts afford accessibility and interpretability, but their discrete, language-mediated nature complicates systematic optimization and limits scalability across heterogeneous tasks. Vector prompts, by contrast, trade explicit semantic grounding for greater flexibility in how control signals are represented and adjusted. Importantly, this distinction concerns the nature of the interface abstraction itself, rather than the particular optimization procedures used to obtain either form of prompt. As such, text and vector prompts should be viewed as fundamentally different customization interfaces, not merely alternative implementations of the same mechanism.

## 2.3. Interface Abstraction vs. Optimization Method

A critical source of confusion in discussions of prompt-based customization is the conflation of interface abstraction with the optimization methods used to instantiate it. These two concerns operate at different conceptual levels and should be clearly separated. The interface abstraction specifies *what kind of control inputs* a model exposes for customization, whereas the optimization method determines *how the values of those inputs are obtained*.

In this work, we treat vector prompts as an interface abstraction rather than as a particular training technique. Exposing vector prompts defines a form of interaction in which users

can condition model behavior through continuous control signals, independent of whether those signals are learned via gradient-based optimization, black-box search, or other inference-only procedures. From this perspective, methods such as prompt tuning (Liu et al., 2022), black-box optimization (Sun et al., 2022b), or future inference-time adaptation techniques are interchangeable mechanisms for populating the same interface, rather than defining the interface itself.

This distinction is especially important in deployment settings where downstream users lack access to model weights, gradients, or training infrastructure. In such cases, the choice of interface abstraction determines the feasible space of customization, while the choice of optimization method reflects practical constraints such as access patterns, query budgets, and latency requirements. Conflating these layers risks attributing limitations of specific optimization procedures to the interface itself, or vice versa.

By separating interface abstraction from optimization method, we emphasize that the central question of this position paper is not how to best optimize prompts, but which forms of control inputs should be exposed to support scalable and inference-only customization. This framing allows empirical analyses to probe the capabilities enabled by a given interface abstraction, while leaving open the development of diverse optimization strategies—particularly black-box and inference-only methods—to realize those capabilities in practice.

Recent work has begun to explore black-box and inference-only approaches to prompt optimization (Sun et al., 2022b; Hu et al., 2024; Sun et al., 2022a), demonstrating that non-gradient methods can meaningfully adapt model behavior when operating over vector prompt representations. While existing black-box techniques generally fall short of the performance achieved by gradient-based prompt tuning (Liu et al., 2022), their effectiveness nonetheless indicates that vector prompts constitute a viable control interface even under strict deployment constraints. From this perspective, gradient-based prompt tuning can be viewed as a diagnostic upper bound on the customization capacity enabled by a given interface abstraction, rather than as the only practical means of realizing that capacity. This suggests that further advances in black-box and inference-only optimization methods may progressively close this gap, reinforcing the importance of exposing vector prompt inputs independently of any specific optimization procedure.

Table 1 organizes existing large language model customization approaches by optimization access and interface layer. By separating interface abstractions from optimization methods, the table clarifies how different approaches align with increasingly restrictive deployment constraints, and highlights inference-only vector-based customization as the most constrained yet practically viable approach.

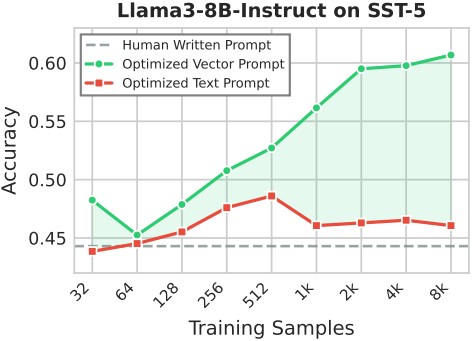

*Figure 2.* Scaling behavior of different prompt interfaces on SST-5 with a fixed LLaMA3-8B Instruct backbone. As the amount of supervision increases, vector-based prompts continue to benefit from additional data, while text-based prompts saturate early. Optimized vector prompts are obtained via gradient-based prompt tuning and serve as a diagnostic upper bound on the customization capacity enabled by vector-based interfaces.

### 2.4. Relation to Representation Engineering

Representation engineering and activation steering methods also control model behavior through continuous directions in representation space (Zou et al., 2023; Rimsky et al., 2024; Li et al., 2023). These methods typically intervene at intermediate layers or internal activations, providing fine-grained control when model internals are accessible. Vector prompt interfaces operate at a different layer of abstraction: they expose continuous control at the input interface, making them more compatible with black-box deployment settings where internal activations are unavailable. We therefore view activation-level steering and input-level vector prompting as complementary: the former studies internal control mechanisms, while the latter provides a deployable interface for downstream customization.

## 3. Empirical Evidence

The empirical analyses in this section are intended as diagnostic evidence rather than a comprehensive benchmark. We use a controlled task setting to isolate differences between prompt interface abstractions, focusing on whether each interface can absorb increasing supervision and how it is utilized during model computation.

### 3.1. Scaling Behavior Under Increasing Supervision

We begin by examining how different prompt interfaces respond to increasing amounts of supervision. Our goal is not to compare optimization algorithms, but to diagnose whether a given interface abstraction can continue to absorb task-specific information as more data becomes available. Accordingly, we fix the backbone model (LLaMA3-

| Train Size | Init Prompt | Text Prompt | Vector Prompt |
|---|---|---|---|
| – | 0.319 | – | – |
| 32 | – | 0.348 | 0.340 |
| 256 | – | 0.310 | 0.370 |
| 2048 | – | 0.313 | 0.379 |
| 8192 | – | 0.344 | 0.401 |

*Table 2.* Instruction-following results on Alpaca using token-level F1. Vector prompts improve steadily as supervision increases, while text prompts remain unstable.

8B Instruct (AI@Meta, 2024)) and task (SST-5 [1]), and vary only the amount of labeled training data used to optimize prompts under different interface choices. Throughout this analysis, gradient-based optimization is used solely as a diagnostic tool to probe interface capacity, rather than as a practical deployment mechanism. Realizing this diagnostic capacity under black-box access remains an open optimization challenge, motivating further work on zeroth-order and inference-only methods tailored to vector prompt interfaces.

Figure 2 shows the resulting scaling behavior. Human-written text prompts provide a stable but limited baseline whose performance is largely insensitive to additional supervision. Optimized text-based prompts (Yuksekgonul et al., 2024) exhibit an initial improvement with small amounts of data, but quickly reach a plateau beyond which additional supervision yields diminishing returns. In contrast, vector-based prompt (Liu et al., 2022) interfaces continue to benefit from increasing supervision across the entire range of training data considered.

We interpret this divergence as evidence of an interface-level bottleneck rather than insufficient optimization. All prompt variants are optimized under comparable conditions, yet only vector-based prompts exhibit sustained scaling behavior. This suggests that the discrete, language-mediated nature of text-based prompts constrains the extent to which additional supervision can be effectively integrated, whereas continuous vector prompts provide a control interface capable of progressively absorbing task-specific information.

Importantly, the vector prompt results in Figure 2 should be understood as a diagnostic upper bound on the customization capacity enabled by vector-based interfaces. The purpose of this experiment is not to advocate gradient-based methods as a deployment solution, but to establish what is achievable when the interface itself does not impose an early saturation point. From this perspective, the observed scaling behavior reflects properties of the interface abstraction rather than the availability of gradient access or the particulars of any optimization procedure.

We also evaluate an instruction-following setting based on Alpaca using token-level F1. As shown in Table 2, the same

[1] https://huggingface.co/datasets/SetFit/sst5

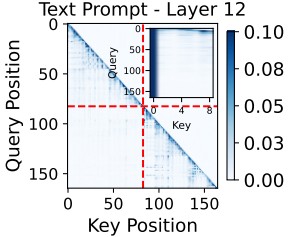
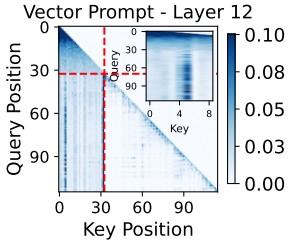

*(a)* Text prompt (Layer 12)     *(b)* Vector prompt (Layer 12)

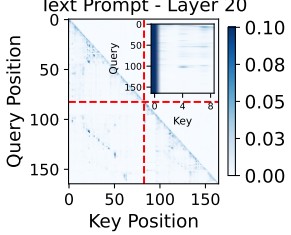
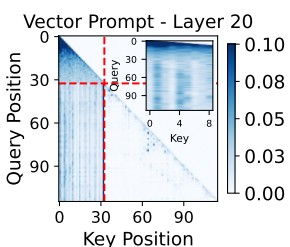

*(c)* Text prompt (Layer 20)     *(d)* Vector prompt (Layer 20)

*Figure 3.* Attention patterns induced by text-based and vector-based prompt interfaces at two representative layers of LLaMA3-8B Instruct model. Layer 12 reflects the integration of prompt information into mid-level representations, while Layer 20 captures the influence of control signals at later stages of computation. Vector prompts exhibit denser and more globally utilized attention across heads, whereas text prompts remain sparse. Similar qualitative differences are observed consistently across layers.

qualitative trend extends beyond sentiment classification: text prompts are unstable across training sizes, whereas vector prompts improve steadily with more supervision.

## 3.2. Mechanistic Differences via Attention Patterns

The scaling behavior observed in Section 3.1 suggests that text-based and vector-based prompt interfaces are incorporated into model computation in fundamentally different ways. To illustrate this distinction, we analyze attention patterns induced by the two interfaces, focusing on how prompt tokens are utilized by task tokens across layers. Our goal is not to explain individual model predictions, or establish a causal account of control, but to provide qualitative evidence that the two interfaces are utilized differently during model computation.

**Experimental Setup.** We visualize attention patterns from LLaMA3-8B Instruct model on the SST-5 task. For vector-based prompting, we prepend 32 learned vector tokens optimized via prompt tuning (Liu et al., 2022). For text-based prompting, we use a discrete prompt optimized using TextGrad (Yuksekgonul et al., 2024). In all cases, prompt tokens are placed as a prefix to the task input. The red dashed line in Figure 3 denotes the boundary between prompt tokens and task tokens, allowing us to examine

whether and how task tokens attend across this interface boundary. We focus on two representative layers: a mid-level layer (Layer 12) and a mid-to-late layer (Layer 20).

**Text Prompts Exhibit Sparse and Non-Distinct Utilization.** For text prompts, attention patterns remain largely segregated across the prompt–task boundary. Prompt tokens do not exhibit utilization patterns that are qualitatively distinct from those of ordinary task tokens: attention to prompt tokens is sparse, localized, and does not systematically increase with depth. Across both Layer 12 and Layer 20, task tokens attend primarily within their local neighborhoods, suggesting that text prompts are incorporated as part of the sequence rather than as dedicated control inputs.

Consistent with prior observations in transformer models, we also observe the persistence of attention sink behavior (Xiao et al., 2023) under text-based prompting. A disproportionate fraction of attention mass remains concentrated on the initial BOS token, indicating that text prompts do not substantially alter the model's default attention allocation dynamics. Together, these patterns suggest that text-based prompts behave as transient, linguistically mediated instructions, whose influence is limited by the same structural constraints governing ordinary sequence tokens.

**Vector-Based Prompts Act as Persistent and Globally Addressable Control Anchors.** Vector-based prompts induce denser utilization by task tokens. Across both examined layers, attention from task tokens consistently crosses the prompt–task boundary, with learned vector prompt tokens being repeatedly and broadly attended to. Unlike text prompts, vector prompts exhibit attention patterns that are stable across layers, indicating that their influence persists throughout the computation. Notably, the presence of vector prompts substantially attenuates attention sink effects. Rather than concentrating attention on a single implicit anchor such as the BOS token, attention is distributed more evenly across the learned prompt tokens. This redistribution suggests that vector prompts provide multiple explicit control anchors, smoothing reliance on default attention sinks and enabling more uniform modulation of model behavior.

**Implications for Interface-Level Control.** These observations point to a qualitative difference in control mechanisms enabled by the two interfaces. Text-based prompts are treated largely as ordinary sequence elements, inheriting sparse utilization patterns and default attention dynamics. Vector-based prompts, by contrast, function as persistent control signals that are globally addressable by task tokens, analogous to learned control modules rather than transient instructions. This utilization difference aligns with the scaling behavior observed in Section 3.1. Interfaces that allow prompt tokens to be persistently and broadly utilized are better suited to absorbing additional supervision, whereas interfaces constrained to linguistically grounded, sparsely utilized tokens are more prone to early saturation. Importantly, these differences arise from the interface abstraction itself, rather than from the specific optimization procedure used to instantiate the prompts.

## 4. Vector Prompts in Deployment

### 4.1. Control Efficiency Under Deployment Constraints

In real-world deployments, downstream application developers typically interact with large language models through text-based prompt interfaces. When such interfaces provide limited control bandwidth, customization is commonly achieved through ad-hoc, sample-driven prompt editing (Do et al., 2024; Murthy et al., 2025): developers iteratively append instructions, constraints, and examples in an attempt to steer model behavior. Over time, this process leads to the accumulation of long, task-specific prompts that are brittle, difficult to maintain, and tightly coupled to particular datasets or usage scenarios.

This growth in prompt length introduces several system-level inefficiencies. As prompts expand, performance may degrade (Liu et al., 2024), where critical control information is diluted by surrounding context. At the same time, longer prompts impose direct computational costs: additional tokens increase inference latency, raise serving costs, and consume context window capacity (Bai et al., 2024) that could otherwise be allocated to task-relevant inputs. Notably, these costs do not arise from task complexity itself, but from the limited efficiency with which text-based interfaces translate supervision into effective control signals.

Our empirical findings suggest that these challenges are rooted in the interface abstraction rather than in prompt engineering practice. As shown in Section 3, text-based prompts exhibit sparse utilization, with their influence largely confined to a small subset of tokens and saturating quickly as supervision increases. In contrast, vector prompt interfaces achieve comparable or superior behavioral control using a small number of learned tokens that are persistently and broadly attended to by task tokens across layers. This difference reflects a substantially higher control efficiency: vector prompts encode task-specific modulation with far fewer tokens, while maintaining influence over model computation.

From a deployment perspective, this efficiency has direct practical implications. By shifting customization from long, linguistically grounded prompts to compact vector-based control signals, developers can avoid prompt explosion while retaining the ability to adapt model behavior across heterogeneous tasks. The resulting customization workflow is more scalable and economical: control inputs remain short, reusable, and amenable to systematic optimization,

even under strict inference-only access constraints. This practical motivation is supported by prior black-box prompt optimization results showing that vector prompts can be optimized under forward-only access (Sun et al., 2022b;a; Hu et al., 2024), together with Figure 2, which shows that the vector interface continues to absorb supervision after optimized text prompts saturate.

## 4.2. Vector Prompts vs. Parameter-Level Customization

It is important to distinguish between parameter-level customization and interface-level control. Existing approaches such as LoRA (Hu et al., 2022) and preference optimization (Rafailov et al., 2023) enable downstream users to modify model parameters in order to achieve task-specific behavior. These methods operate directly on model weights and are well suited for a small number of stable, high-value tasks where training, evaluation, and redeployment costs can be amortized over long-term use.

However, in deployed LLM ecosystems, customization demands are rarely confined to a few static tasks. Downstream applications often involve a large and continuously evolving set of objectives, with frequent updates driven by changing data distributions, user requirements, and product constraints. Under these conditions, relying primarily on parameter-level customization requires repeated training and deployment cycles, which impose substantial operational overhead. These include increased training and evaluation costs, complex model version management, and ongoing maintenance burdens that scale poorly as the number of supported tasks grows.

Vector prompt interfaces offer an alternative mode of customization that operates at a different layer. Rather than modifying model parameters, vector prompts condition a shared base model through compact control inputs applied entirely at inference time. This separation allows task-specific behavior to be expressed without retraining or redeploying the model, enabling multiple customization targets to coexist over a single deployed backbone. From an architectural perspective, vector prompts decouple rapid, fine-grained adaptation from the slower and more expensive processes associated with parameter updates.

Importantly, this alternative does not render parameter-level customization obsolete. Fine-tuning remains an appropriate mechanism for coarse or infrequent adaptation, such as aligning a model to a new domain or distribution. Vector prompt interfaces provide a more scalable and operationally efficient abstraction for frequent and rapidly evolving tasks. By enabling customization to remain at the level of control inputs rather than model weights, vector prompt interfaces reduce deployment complexity while preserving flexibility in real-world systems. At the same time, this flexibility introduces management overhead: systems may need to store, validate, version, and debug many task- or user-specific control vectors. Thus, vector prompt interfaces should complement rather than replace text prompts and parameter-level adaptation, especially in settings where semantic inspectability or centralized training simplicity is more important than rapid per-task customization.

## 4.3. Inference-Only Constraints

In practice, most downstream users interact with large language models under inference-only access constraints. Model weights, gradients, and activation access are typically unavailable, either due to platform restrictions, operational complexity, or cost considerations. As a result, customization mechanisms that assume gradient access or repeated training are often impractical outside of a small number of centralized workflows. This constraint is becoming more pronounced as models and deployment settings scale. Increasing context lengths (Qiu et al., 2025) and larger model sizes (Yang et al., 2025) amplify the memory and compute costs associated with gradient-based adaptation, particularly due to the need to store activations during backpropagation. Even when fine-tuning is technically supported, the overhead of provisioning training hardware, managing activation memory, and coordinating retraining cycles can exceed what is feasible for rapid, task-specific iteration. In contrast, inference-time adaptation avoids these costs entirely, operating within the same execution path as standard model serving.

From a system-level perspective, these trends favor customization mechanisms that are compatible with inference-only workflows. Vector prompt interfaces align naturally with this requirement: they introduce no additional training-time dependencies, do not require access to intermediate activations, and can be applied using the same infrastructure used for standard inference. This compatibility enables downstream users to adapt model behavior while relying solely on inference resources, such as serving-optimized hardware and inference-oriented software stacks.

Taken together, these considerations suggest that inference-only customization is not merely a convenience, but an increasingly necessary design constraint. As models grow larger and contexts longer, interfaces that rely on parameter updates or gradient access become progressively less viable for frequent customization. By exposing vector prompt interfaces, model providers can support adaptation strategies that remain efficient, scalable, and compatible with real-world system constraints, without entangling customization with the escalating costs of training-time computation.

# 5. Security and Risk Considerations

Exposing new control interfaces for large language models naturally raises concerns about security and information leakage. These concerns help explain why current providers may be reluctant to expose such interfaces: continuous inputs are less semantically inspectable than text, may bypass text-level safety filters, and could interact with provider-side control mechanisms in unintended ways. At the same time, these concerns highlight an important research gap: provider-facing vector interfaces require systematic study of safety, controllability, and co-design between model providers and downstream developers. Our position aims to motivate such work so that customization interfaces can evolve together with provider-side safeguards and deployment requirements. In this section, we clarify the threat model under which our position applies and articulate why exposing vector prompt inputs does not introduce a fundamentally new attack surface relative to existing text-based prompting interfaces. Our argument is grounded in standard information-theoretic reasoning about observability and access constraints, rather than claims of absolute security or a complete empirical security evaluation.

## 5.1. Threat Model

We consider a standard black-box deployment setting that reflects the access patterns of downstream users in practice. Attackers are assumed to have query access to the model and can observe only its outputs. They do not have access to model parameters, internal activations, gradients or intermediate computation states. Query budgets, rate limits, and monitoring mechanisms are assumed to be in place, as is typical for deployed large-scale LLM services. This threat model intentionally matches that of existing text-based prompting APIs. We do not consider white-box attacks or scenarios in which internal model states are directly exposed, as such settings fall outside the scope of current commercial and enterprise deployments.

## 5.2. An Information-Theoretic Perspective

Under black-box access, information leakage from a deployed model is fundamentally constrained by what is observable through the output channel and by the number of allowed queries. Regardless of the form of the input—whether discrete text tokens or continuous vector prompts—an attacker can only extract information that manifests in generated outputs. From an information-theoretic perspective, exposing vector prompt inputs does not increase the capacity of this observable channel. The space of possible outputs, the sampling mechanisms, and the mapping from inputs to outputs remain unchanged. While different input parameterizations may alter how efficiently specific behaviors are elicited, they do not expand the set of information that can be observed in principle. As a result, the potential leakage remains bounded by the same output distributions and access constraints that apply to text-based prompting.

We emphasize that we do not claim a formal proof of risk equivalence, as such guarantees are generally unattainable under adaptive black-box access. Our argument relies on the standard distinction between search efficiency and information capacity: interfaces may differ in how efficiently they explore a model's behavior space, without altering the fundamental limits of what can be observed through outputs. Accordingly, our position is not that vector prompt interfaces eliminate security risk, but that under black-box access they do not introduce a qualitatively new form of observability beyond existing text-based prompt interfaces. Empirically characterizing how different prompt interfaces affect attack efficiency remains an important open problem for future work. In practice, exposing vector inputs should therefore be accompanied by interface-level safeguards, such as constraining allowable vector norms or subspaces, monitoring abnormal query patterns, and limiting how vector prompts can be queried or adapted.

# 6. Alternative Views

## 6.1. Text Prompts Already Enable Customization

One alternative view holds that text-based prompting already provides a flexible and effective mechanism for customizing large language models. In practice, a wide range of applications are successfully built by carefully crafting instructions, constraints, and examples in natural language, suggesting that additional control interfaces may be unnecessary. This perspective is reasonable in settings where customization is infrequent, tasks are relatively stable, and prompt authorship can be handled manually. Text prompts are human-readable and easy to deploy, making them well suited for rapid prototyping and low-overhead adaptation.

However, this view conflates usability with scalability. As customization demands grow in frequency and diversity, text prompts tend to become long, brittle, and difficult to maintain, and their effectiveness often saturates under systematic optimization. Our position is not that text prompts fail to customize models, but that they are ill-suited as a scalable control interface for inference-only customization under realistic deployment constraints.

## 6.2. Exposing Vector Prompt Increases Security Risk

A second alternative view argues that exposing vector prompt inputs may increase the risk of information leakage. From this perspective, allowing continuous control signals could enable more precise probing of model behavior and potentially amplify existing security concerns.

This concern is well founded, as no prompt-based interface is free of risk, and deployed models must already contend with adversarial prompting and systematic behavior exploration. Caution is therefore warranted when proposing new forms of control access. Nevertheless, under standard black-box threat models, the relevant security question is not whether risk exists, but whether a new interface introduces a qualitatively different attack surface. As discussed in Section 5, vector prompt interfaces do not alter the observable output channel or expose internal model states. They may affect the efficiency with which behaviors are discovered, but do not increase the fundamental information available through outputs. Accordingly, our position is that vector prompt interfaces are risk-equivalent to text-based prompting under comparable access constraints.

### 6.3. Vector Prompts May Complicate Monitoring and Debugging

Another concern is that vector prompt interfaces may make monitoring and debugging more difficult than text prompts. Text prompts can often be inspected, filtered, or rewritten using semantic tools, whereas vector prompts are not directly human-readable. This opacity may complicate safety monitoring, prompt-injection defenses, and diagnosis of user- or task-specific failures. We view this as an important trade-off of the proposed interface: vector prompts can provide more efficient control, but they require new monitoring strategies and interface-level safeguards rather than relying solely on textual inspection.

### 6.4. Optimization Detail vs. Interface Design

A third alternative view holds that the distinction between text and vector prompts reflects an optimization choice rather than an interface-level concern. From this standpoint, improvements attributed to vector prompts could be seen as artifacts of gradient-based tuning, rather than evidence for a fundamentally different form of control.

This interpretation is natural given that much prior work on vector prompts has focused on optimization techniques. If vector prompts are viewed solely as a parameterization optimized by specific algorithms, their relevance to deployed, inference-only systems may appear limited. Our position differs by separating interface abstraction from optimization method. The interface determines what form of control signals a model accepts and what kinds of supervision can be absorbed, independent of how those signals are obtained. From this perspective, the distinction between text-based and vector-based prompts is not merely an implementation detail, but a choice that shapes the controllability, scalability, and system-level feasibility of customization.

## 7. Call to Action

**For model providers**, we argue that vector prompt inputs should be elevated from an internal optimization mechanism to a customization interface. This entails exposing constrained vector prompt APIs that allow downstream users to inject, reuse, and manage fixed-length control vectors at the input-encoding stage, under the same black-box access assumptions as existing text-based prompting. Crucially, such interfaces should decouple customization from model parameters, enabling systematic inference-only adaptation without exposing weights, gradients, or internal activations.

**For the research community**, we call for a shift in emphasis from improving individual prompt-tuning algorithms to characterizing prompt interfaces as control abstractions. This includes developing better inference-only and black-box methods that operate effectively over vector prompt inputs, as well as evaluation frameworks that assess control efficiency, scalability under increasing supervision, and behavioral stability across tasks and iterations, rather than focusing solely on task-level performance.

**For system builders and practitioners**, we argue that customization workflows should shift from ad hoc, manual editing of text prompts to data-driven optimization. Rather than repeatedly modifying textual instructions to satisfy immediate objectives—often at the expense of long-term maintainability— practitioners should construct task-specific training and evaluation datasets that enable systematic optimization of vector prompt inputs. By treating customization as a data-driven process, vector prompts can be iteratively improved, evaluated, and reused across tasks, reducing reliance on brittle prompt engineering while making customization behavior more stable and maintainable over time under inference-only deployment constraints.

## 8. Conclusion

This position paper argues that customization has become a central bottleneck for scalable LLM deployment. By re-framing prompting as an interface abstraction, we identify vector prompt inputs as a compact and flexible means of customizing model behavior without modifying model parameters. Our empirical and system-level analyses position vector prompts as a complementary control interface to text-based prompting under inference-only deployment constraints. This perspective does not diminish the value of text prompts or fine-tuning; rather, it highlights the need for complementary control surfaces that make customization more systematic, optimizable, and maintainable in deployed LLM systems.

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
