# OpenReview forum: "Position: Vector Prompt Interfaces Should Be Exposed to Enable Customization of Large Language Models"
_ICML.cc/2026/Position_Paper_Track — ICML 2026 Position Paper Track regular_

### Official Review · Reviewer_xiTT · 2026-03-09

**Significance:** 3
**Argument Clarity:** 3
**Rating:** 4
**Confidence:** 3

**Questions:**

Q1. In the core comparison of Figure 2, the vector prompt uses gradient-based optimization while the text prompt uses TextGrad. In the black-box deployment scenario advocated in the paper, how large is the performance gap between the vector prompt optimized by gradient-free methods such as BBT and the text prompt optimized by TextGrad? If they are close, what is the actual gain of exposing the vector interface?

Q2. Currently, no mainstream model provider exposes a vector prompt API. Are the authors aware of the specific concerns on the provider side? In particular, is it possible that user-injected continuous vectors might interfere with or conflict with the provider's internal safety alignment mechanism?

Q3. Recent work on representation engineering and activation steering shows that model behavior can be precisely controlled by injecting direction vectors in intermediate layers. What is the relationship between these methods and the input-level vector prompt proposed in the paper? Why didn't the paper discuss this related work?

Q4. The experiments were only conducted on SST-5 (sentiment classification). Can this scaling behavior be reproduced on more complex tasks (such as code generation, multi-turn dialogue, and instruction following)?

**Alternative Views Section:**

Yes

**Compliance With Llm Reviewing Policy A Conservative:**

Affirmed.

**Discussion Potential:**

3

**Final Justification:**

My concerns are adequately addressed, including the additional Alpaca experiment. I maintain my score.

**Paper Summary:**

This paper argues that model providers should expose vector prompt inputs as part of the public interface for LLM customization. The position is supported by a scaling experiment on SST-5 and attention pattern analysis, alongside discussions of deployment efficiency and security risk under black-box access.

**Position:**

Yes

**Position In Title:**

Yes

**Related Work:**

3

**Strengths And Weaknesses:**

S1. Redefining prompting as "interface abstraction" rather than "optimization technique" is the paper's greatest contribution. This reframing elevates the discussion from "which prompt tuning algorithm is better" to "what form of control interface should the model provider expose," representing a system design-level consideration that is enlightening for the ICML community.

S2. The classification system in Table 1 is clear and useful. Organizing existing custom methods along the dimensions of optimization access and interface layer, it places fine-tuning, prompt tuning, text prompt optimization, and black-box vector optimization within a unified framework, helping readers understand the positioning and trade-offs of different methods.

S3. The framing of the security discussion is reasonable. The paper does not shy away from security issues but argues from an information-theoretic perspective: under the black-box setting, vector prompting does not change the information capacity of the output channel, thus not introducing a fundamentally new attack surface. Although this argument has limitations, it is appropriate for a position paper.

S4. The scope is clearly defined. The paper explicitly states that it is not proposing a new algorithm, does not claim that vector prompts are superior to text prompts on all tasks, and does not argue for the elimination of fine-tuning. This self-aware scope limitation is a plus in positional papers.


----

W1. The experimental evidence is weak, and the core comparisons raise fairness issues. Only one task (SST-5) and one model (LLaMA3-8B) are considered. More importantly, Figure 2 shows that the vector prompt uses gradient-based prompt tuning, while the text prompt uses TextGrad—the optimizers are completely different. The paper explains this design choice as "diagnostic upper bound," but this framing itself avoids the truly crucial question: in real-world black-box scenarios, how do vector prompts optimized by gradient-free methods like BBT compare to text prompts optimized by TextGrad? If the difference is small, the actual value of exposing the vector interface needs to be re-evaluated.

W2. The paper doesn't seriously discuss why providers don't expose the vector interface. The paper calls for providers to open up the vector prompt API, but currently no mainstream providers do so, which is a phenomenon that needs explanation. The paper's security discussion only extends to the abstract information-theoretic level, neglecting concrete issues that the provider might genuinely care about: for example, continuous vectors potentially bypassing text-level safety filters, user-injected vectors interfering with the provider's internal alignment control vectors, and exposing the embedding-level interface indirectly exposing the model's internal representation structure.

W3. There is a lack of discussion on work related to representation engineering/steering vectors. Recent work has extensively explored controlling model behavior through intermediate-layer direction vectors (activation steering, etc.), which also operate in the continuous space and demonstrate significant effectiveness in behavior control. The paper completely lacks any citation or discussion of this line of work, a clear deficiency in related research. The relationship between these methods and the input-level vector prompt (substitution, complementarity, or competition) needs clarification.

W4. The explanatory power of the attention pattern analysis is limited. Section 3.2 observes that vector prompts induce more dense and global attention patterns and mitigate the attention sink phenomenon. However, attention sinking is a structural phenomenon of transformers. Redistributing attention from BOS tokens to learned tokens does not necessarily mean better "control"—it may simply be changing the location of attention concentration. This mechanical claim requires more grounded analysis to support it.

**Support:**

2

---

> ### Author Rebuttal · Authors · 2026-03-31
>
> We thank the reviewer for the helpful and detailed feedback. Our rebuttals are organized as follows.
>
> ---
>
> ## W1 and Q1. Fairness of comparison and black-box optimization gap
>
> We thank the reviewer for this important question.
>
> We agree that Fig. 2 is not a fully matched black-box comparison (gradient-based vs. TextGrad). Our goal is not to compare optimization methods, but to use this setup as a **diagnostic experiment** to study interface capacity.
>
> Regarding Q1, current gradient-free methods may not yet fully exploit high-dimensional vector prompts and can be comparable to text-based optimization in practice. However, prior work (e.g., [1]) shows that black-box optimization of continuous prompts can approach gradient-based performance in smaller settings, suggesting the gap is largely methodological.
>
> The key insight of Fig. 2 is the distinction between **interface capacity and optimization capability**: vector-based interfaces continue to improve with more supervision, while text prompts tend to saturate early. This is further supported by recent work (e.g., TokMem [2], Gist Tokens [3]) showing that long textual context can be compressed into compact vector representations with strong performance.
>
> Therefore, the limitation lies less in the representation itself and more in the lack of accessible interfaces for studying and optimizing vector-based control in black-box settings. Exposing such interfaces enables more expressive control and better utilization of this capacity.
>
>
> [1] "Black-box tuning for language-model-as-a-service." In International Conference on Machine Learning, pp. 20841-20855. PMLR, 2022.
>
> [2] "TokMem: One-Token Procedural Memory for Large Language Models." In The Fourteenth International Conference on Learning Representations.
>
> [3] "Learning to compress prompts with gist tokens." Advances in Neural Information Processing Systems 36 (2023): 19327-19352.
>
> ---
>
> ## W2 and Q2. Security and adversarial risks
>
> We thank the reviewer for raising this important question.
>
> We agree that vector-based interfaces are not exposed, likely due to concerns about **security, controllability, and system complexity**. Compared to text, vector inputs are less interpretable, harder to monitor, and may enable more efficient exploration of model behaviors.
>
> However, we argue these challenges are **manageable through interface design**, similar to existing text-based APIs. Possible safeguards include:
> - **Input constraints** (e.g., bounded regions or subspaces),
> - **Monitoring and filtering** (e.g., anomaly detection),
> - **Controlled abstraction** (e.g., structured parameters instead of raw embeddings).
>
> More broadly, we view this as a gap between current interfaces and the evolving needs of real-world model customization. Exposing vector-based interfaces enables more expressive control and supports systematic study of both optimization and safety under realistic settings.
>
> We will revise the paper to clarify these points.
>
> ---
>
> ## W3 and Q3. Relation to representation engineering and activation steering
>
> We thank the reviewer for pointing out this line of work and apologize for not discussing it.
>
> We agree that activation steering and related methods also enable behavior control via **continuous representations**. The key difference is the level of intervention: activation steering operates on internal layers, while vector prompts act at the input interface, making them more suitable for black-box settings.
>
> We view these approaches as **complementary**: internal methods provide fine-grained control with model access, while input-level interfaces offer a more deployable control mechanism. We will revise the paper to clarify this connection.
>
> ---
>
> ## W4 and Q4. Attention analysis and generalization
>
> We thank the reviewer for the helpful comments.
>
> For W4, we agree that attention patterns do not establish causal control. Our analysis is intended as a **supporting observation** that the calculation mechanism of text/vector prompt is totally different, and we will revise the text to avoid over-interpretation.
>
> For Q4, we conducted additional experiments on an instruction-following task (Alpaca, Token F1 is the metrics) and observe a similar trend as Fig. 2:
>
> | Train Size | Init Prompt | TextGrad  | Soft vector tuning |
> |-----------|-------------|-------------|---------|
> | –         | 0.319       | –           | –       |
> | 32        | –           | 0.348       | 0.340   |
> | 256       | –           | 0.310       | 0.370   |
> | 2048      | –           | 0.313       | 0.379   |
> | 8192      | –           | 0.344       | 0.401   |
>
> TextGrad is unstable and does not consistently outperform the initial prompt, while soft vector tuning shows steady improvements with more data.
>
> This suggests the scaling behavior generalizes beyond SST-5, supporting our claim on the advantage of vector-based interfaces. We will include full results in the revised version.

---

> > ### Author Rebuttal · Reviewer_xiTT · 2026-04-05
> >
> > Thanks for the thorough response and the additional Alpaca experiment! My concerns are adequately addressed. I am maintaining my positive score and look forward to seeing the revisions incorporated.

---

### Official Review · Reviewer_1hhu · 2026-03-12

**Significance:** 3
**Argument Clarity:** 3
**Rating:** 5
**Confidence:** 4

**Questions:**

See weaknesses.

**Alternative Views Section:**

Yes

**Compliance With Llm Reviewing Policy A Conservative:**

Affirmed.

**Discussion Potential:**

3

**Paper Summary:**

The paper argues that vector prompts interfaces provides a more flexible interface for customizing model behavior at inference time.

Recent works such as GEPA [1] demonstrate that searching for improved prompts provides improvements comparable to fine-tuning in several instances. However, doing so at inference time without outcome supervision can be infeasible. The paper’s position on vector prompt interfaces would obviate the need for such search procedures, particularly based on gradient based learning approaches.

The authors argue the position well and provide evidence regarding the greater flexibility of prompt vector interfaces, compared to textual prompts. Incorporating prompt vector interfaces and exposing them to users could potentially even offer greater customization than prompt based optimization, and by citing relevant works and demonstrating the ease of optimization of such interfaces the authors provide evidence to support their position. The authors also consider relevant alternate views, particularly regarding the risk posed by such interfaces.

**Position:**

Yes

**Position In Title:**

Yes

**Related Work:**

3

**Strengths And Weaknesses:**

1. The paper is clearly written and presents a well-structured argument for exposing vector prompt interfaces as part of LLM customization pipelines.
2. The authors carefully position vector prompt interfaces as complementary to existing techniques such as fine-tuning and prompt optimization rather than as replacements.
3. The discussion of alternative viewpoints strengthens the paper and demonstrates awareness of potential concerns around security and system design.

Some alternate views that could have been considered and made the position stronger:
1. A significant weakness of such interfaces, as acknowledged by the authors themselves, is the risk of such systems. While input classifiers, such as constitutional classifiers, have shown some promise in mitigating the risk posed by language models and prompt injections. Moving to even more black-box methods, such as vector prompt interfaces, might make monitoring harder.
2. Additionally, such interfaces, particularly when conditioned on specific users might make it fine-tuning, debugging and any monitoring more cumbersome. For instance, when training new models or fine-tuning existing models, the flexibility provided by text only prompts makes training simpler, requiring user specific prompts can increase the cost of training.
3. For a number of reasoning models and interfaces for generative models, such as diffusion models, etc. Several current interfaces edit the user provided prompt. For example, see MovieGen [1]. The authors do not adequately discuss such optimizations.


[1] Polyak, Adam, et al. "Movie gen: A cast of media foundation models." arXiv preprint arXiv:2410.13720 (2024).

**Support:**

3

---

> ### Author Rebuttal · Authors · 2026-03-28
>
> We thank the reviewer for the positive feedback and for the insightful suggestions, particularly the additional alternative views that can help make the paper more comprehensive. We address the points below.
>
> ---
>
> ## On security, monitoring, and debugging challenges
>
> We agree that vector prompt interfaces may introduce additional challenges for monitoring, debugging, and safety control, particularly compared to text-based interfaces where semantic inspection is more direct.
>
> We will revise the paper to more explicitly discuss these trade-offs, including the potential need for new monitoring strategies and interface-level safeguards when moving toward vector-based customization.
>
> ---
>
> ## On training and system complexity
>
> We appreciate this point and agree that conditioning on user-specific vector prompts may introduce additional complexity in training and system maintenance.
>
> We will revise the paper to better clarify how vector prompt interfaces are intended to complement, rather than replace, existing approaches such as fine-tuning and prompt-based methods, and to discuss the associated system-level trade-offs more explicitly.
>
> ---
>
> ## On related work (e.g., prompt editing and MovieGen)
>
> We thank the reviewer for pointing out this relevant direction. We agree that systems which modify or optimize user prompts (e.g., MovieGen) are closely related in spirit.
>
> We will revise the paper to incorporate and discuss such approaches (e.g., MovieGen), and clarify how they relate to our position on interface design for model customization.

---

> > ### Author Rebuttal · Reviewer_1hhu · 2026-04-03
> >
> > Thanks for the response! I am maintaining my positive score.

---

### Official Review · Reviewer_Lra7 · 2026-03-13

**Significance:** 3
**Argument Clarity:** 3
**Rating:** 4
**Confidence:** 4

**Questions:**

How do you propose mitigating the severe security and jailbreak vulnerabilities that would inevitably arise from granting external users direct access to the continuous vector space of close-source models, for example, via a API?

**Alternative Views Section:**

Yes

**Compliance With Llm Reviewing Policy A Conservative:**

Affirmed.

**Discussion Potential:**

3

**Final Justification:**

Main concerns addressed. The paper itself is a good paper which deserves to be presented in ICML.

**Paper Summary:**

The paper argues for the postition that pure text prompts are insufficient for scalable and stable LLM customization. It calls for model providers to expose continuous vector prompt interfaces to allow for deeper control under black-box inference constraints.

**Position:**

Yes

**Position In Title:**

Yes

**Related Work:**

4

**Strengths And Weaknesses:**

Pros:

1. The paper accurately points out the limitations of text prompts (discreteness and semantic dependency) in complex customization and continuous learning tasks.

Cons:

I don't have a major concern about the paper, several minor concerns mainly lie on:

1. The experimental section is extremely thin, relying only on small-scale validation on the basic SST-5 sentiment classification dataset, making it unconvincing.

2. The potential adversarial risks. This can be associated with opening underlying vector interfaces.

3. Formatting: Please do not forget the running title.

**Support:**

3

---

> ### Author Rebuttal · Authors · 2026-03-28
>
> We thank the reviewer for the positive feedback and for recognizing the motivation and potential of this work. We address the concerns below.
>
> ---
>
> ### On the experimental scope
>
> We thank the reviewer for this feedback and agree that the current experimental setup is limited in scale.
>
> Our intention is not to provide a comprehensive benchmark, but to present **diagnostic evidence** illustrating fundamental differences between text-based and vector-based interfaces. In particular, the results on SST-5, together with the scaling behavior shown in Fig. 2, highlight that vector prompts can continue to absorb supervision, whereas text prompts exhibit early saturation.
>
> We will revise the paper to better clarify this goal and more explicitly connect these observations to practical implications under deployment constraints.
>
> ---
>
> ### On security and adversarial risks
>
> We agree that exposing vector interfaces raises important security considerations.
>
> As discussed in Sec. 5, our argument is not that such interfaces are risk-free. While exposing vector inputs may introduce new ways of interacting with the model, these interactions remain mediated through the model’s output channel in black-box settings. We therefore view the primary difference as affecting how efficiently model behaviors can be explored, rather than fundamentally changing the nature of access.
>
> More broadly, we view the potential risks and attack surfaces introduced by vector-based interfaces as an important and timely research direction. We hope this position paper can help draw attention to this question and encourage systematic investigation into safer and more effective interface designs for deploying LLMs in real-world settings.
>
> We will revise the paper to better clarify this scope and explicitly discuss the associated risks.
>
> ---
>
> ### On potential vulnerabilities with API exposure
>
> We thank the reviewer for this important question.
>
> Our position is that exposing vector inputs does not fundamentally change the nature of access compared to existing text-based APIs, as both operate through a constrained output channel in black-box settings. However, we agree that vector interfaces may enable more efficient exploration of model behaviors, which could amplify certain risks.
>
> To mitigate such risks, we envision several possible directions at the interface level, including: (1) constraining the allowable vector space (e.g., via norm bounds or subspace restrictions), (2) incorporating monitoring mechanisms to detect abnormal usage patterns or outputs, and (3) designing controlled access protocols that limit how vector inputs can be queried or adapted. And we believe studies in these directions will also foster our understanding of LLM computing mechanism.
>
> We will revise the paper to better discuss these potential mitigation directions and clarify the associated design considerations.
>
> ---
>
> ### On formatting
>
> We thank the reviewer for pointing this out and will correct the running title in the revision.

---

> > ### Author Rebuttal · Reviewer_Lra7 · 2026-04-02
> >
> > Thank you for the response. I will maintain my positive rating.

---

### Official Review · Reviewer_adKZ · 2026-03-14

**Significance:** 2
**Argument Clarity:** 2
**Rating:** 4
**Confidence:** 2

**Questions:**

None

**Alternative Views Section:**

Yes

**Compliance With Llm Reviewing Policy A Conservative:**

Affirmed.

**Discussion Potential:**

2

**Final Justification:**

Given the author's response, I understood the main novelty of this work, and agree that their claims are supported. I'm raising my score.

**Paper Summary:**

This paper argues that the "vector prompt" instead of "text prompt" is useful for scalable, stable and inference-only customization in LLMs. Thus, the authors claim that the model providers should expose the vector prompt inputs in the public interface.

**Position:**

Yes

**Position In Title:**

Yes

**Related Work:**

3

**Strengths And Weaknesses:**

Strength
- This paper focuses on the important problem of customizing LLMs.
- Observation on the scalability of vector prompt and text prompt is meaningful.

Weakness
- It is unclear whether suggesting the interface abstraction is novel, given that related vector prompt methods are already proposed in the literature.
- I am little concerned with the security issue of the proposed interface. Sec.5 does not have any experimental results.
- No empirical results supporting why the proposed interface is useful in practice.

---
Post-rebuttal comment: given the author's response, my concerns are resolved.

**Support:**

2

---

> ### Author Rebuttal · Authors · 2026-03-27
>
> We thank the reviewer for the thoughtful feedback. Our rebuttal is provided point by point as follows:
>
> ---
>
>
> ## On Novelty of the Interface Abstraction
>
> We thank the reviewer for this point and agree that prior work has explored vector prompt methods (e.g., [1,2]).
>
> Our contribution lies at the **level of interface design**, rather than introducing a new optimization method. Existing work treats vector prompts as internal optimization techniques, whereas we argue they should be **exposed as part of the public interface** to enable systematic and inference-only customization under deployment constraints.
>
> From this perspective, prior work does not diminish novelty, but instead provides supporting evidence that such an interface is viable and expressive. We will revise the paper to make this distinction more explicit and avoid potential misunderstanding.
>
> ---
>
> ## On Security Concerns
>
> We thank the reviewer for raising this important concern and agree that security considerations are critical.
>
> Our goal in Sec. 5 is not to provide a full empirical evaluation of security, but to clarify the **threat model**. Specifically, we consider a standard black-box setting where only model outputs are observable. Under this setting, security risks are fundamentally constrained by the output channel, and our argument is that vector prompt interfaces do not introduce a **qualitatively new attack surface**, but may affect the efficiency of behavior exploration.
>
> More broadly, we view the question of *how different interfaces affect security risk* as an important and currently underexplored issue that warrants further investigation by the community, rather than one that can be conclusively resolved within a single position paper.
>
> We will revise the section to better emphasize this scope and clarify the limitations of our claims.
>
> ---
>
> ## On Practical Usefulness and Empirical Evidence
>
> We thank the reviewer for this important point and agree that demonstrating practical usefulness is essential.
>
> Our goal is not to provide a full system-level evaluation, but to present **evidence and signals** indicating that vector-based interfaces are practically useful under deployment constraints. Prior work has shown that vector prompts can be effectively optimized even in **forward-only / black-box settings**, achieving strong performance without gradient access (e.g., [1,2,3]). These results suggest that vector-based control signals are both expressive and usable under realistic access constraints.
>
> In our paper, Fig. 2 provides complementary evidence from an interface perspective: text-based prompts exhibit early saturation as supervision increases, whereas vector prompts continue to improve, indicating a stronger ability to absorb task-specific information. Together, these observations suggest that exposing vector prompt inputs can enable more effective and scalable customization compared to text-only interfaces.
>
> We will revise the paper to better connect these empirical observations and prior results to practical system-level implications.
>
> ---
>
> Hope we have addressed all of your concerns. Let us know if you have any further questions :)
>
> ---
>
> [1] Sun, Tianxiang, Yunfan Shao, Hong Qian, Xuanjing Huang, and Xipeng Qiu. "Black-box tuning for language-model-as-a-service." In International Conference on Machine Learning, pp. 20841-20855. PMLR, 2022.
>
> [2] Yu, Lang, Qin Chen, Jiaju Lin, and Liang He. "Black-box Prompt Tuning for Vision-Language Model as a Service." In IJCAI, pp. 1686-1694. 2023.
>
> [3] Sun, Tianxiang, Zhengfu He, Hong Qian, Yunhua Zhou, Xuan-Jing Huang, and Xipeng Qiu. "Bbtv2: Towards a gradient-free future with large language models." In Proceedings of the 2022 conference on empirical methods in natural language processing, pp. 3916-3930. 2022.

---

> > ### Author Rebuttal · Reviewer_adKZ · 2026-04-03
> >
> > I thank the authors for the clarification. It seems like I had some mis-interpretation on the novelty & position of this paper. My concerns are resolved, and I will raise my score.

---

### Decision · Program_Chairs · 2026-04-30

**Decision:**

Accept (regular)

**Comment:**

The submission received consistently positive scores (4/4/4/5), with reviewers praising the paper for its "enlightening" reframing of prompting as an interface abstraction rather than merely an optimization technique. While reviewers initially raised valid concerns regarding e.g. the narrow experimental scope, under-explored security risks, and the lack of discussion on practical system complexities, the authors successfully resolved these issues during the rebuttal period. Overall, this paper highlights a critical bottleneck in scalable LLM customization and makes a solid case for vector prompt interfaces, which are indeed not common as of today. As a result, I am recommending acceptance, provided the authors include their promised rebuttal updates in the final version.